# Integrated Transcriptomic, Proteomic, and Metabolomics Analysis Reveals Peel Ripening of Harvested Banana under Natural Condition

**DOI:** 10.3390/biom9050167

**Published:** 2019-04-30

**Authors:** Ze Yun, Taotao Li, Huijun Gao, Hong Zhu, Vijai Kumar Gupta, Yueming Jiang, Xuewu Duan

**Affiliations:** 1South China Botanical Garden, Chinese Academy of Sciences, Guangzhou 510650, China; yunze@scbg.ac.cn (Z.Y.); litaotao@scbg.ac.cn (T.L.); zhuhong@scbg.ac.cn (H.Z.); ymjiang@scbg.ac.cn (Y.J.); 2Institute of Fruit Tree Research, Guangdong Academy of Agricultural Sciences, Guangzhou 510640, China; huijun_gao@aliyun.com; 3Department of Chemistry and Biotechnology, ERA Chair of Green Chemistry, School of Science, Tallinn University of Technology, 12618 Tallinn, Estonia; vijai.gupta@ttu.ee

**Keywords:** auxin, banana, fruit ripening, metabolomics, proteomics, transcriptomics

## Abstract

Harvested banana ripening is a complex physiological and biochemical process, and there are existing differences in the regulation of ripening between the pulp and peel. However, the underlying molecular mechanisms governing peel ripening are still not well understood. In this study, we performed a combination of transcriptomic, proteomic, and metabolomics analysis on peel during banana fruit ripening. It was found that 5784 genes, 94 proteins, and 133 metabolites were differentially expressed or accumulated in peel during banana ripening. Those genes and proteins were linked to ripening-related processes, including transcriptional regulation, hormone signaling, cell wall modification, aroma synthesis, protein modification, and energy metabolism. The differentially expressed transcriptional factors were mainly ethylene response factor (ERF) and basic helix-loop-helix (bHLH) family members. Moreover, a great number of auxin signaling-related genes were up-regulated, and exogenous 3-indoleacetic acid (IAA) treatment accelerated banana fruit ripening and up-regulated the expression of many ripening-related genes, suggesting that auxin participates in the regulation of banana peel ripening. In addition, xyloglucan endotransglucosylase/hydrolase (XTH) family members play an important role in peel softening. Both heat shock proteins (Hsps) mediated-protein modification, and ubiqutin-protesome system-mediated protein degradation was involved in peel ripening. Furthermore, anaerobic respiration might predominate in energy metabolism in peel during banana ripening. Taken together, our study highlights a better understanding of the mechanism underlying banana peel ripening and provides a new clue for further dissection of specific gene functions.

## 1. Introduction

Banana is a major staple food and export product in many countries, with an annual output of 102 million tons worldwide (http://faostat.fao.org). Banana is a typical climacteric fruit with a pre-climacteric phase followed by a peak in the ethylene production that co-ordinates a series of ripening-associated processes, including climacteric respiration, pulp softening, peel de-greening, and aroma production [1]. Once the ripening is initiated, the processes are irreversible and rapid, leading to a short shelf life of three to five days. Ripening also decreases the fruit resistance to both mechanical damage and microbial infection [2], which usually causes enormous economic losses. Therefore, a better understanding of the ripening attributes of banana fruit may help to develop strategies to improve the nutritional and sensorial quality and reduce the postharvest losses of the fruit.

In the past decades, numerous studies have been performed to investigate the mechanism involved in banana fruit ripening at physiological, biochemical, and molecular levels. Most studies have been focused on ethylene biosynthesis, perception, and signaling [2,3]. Recently, studies on the transcriptional regulation of ripening-related genes have deepened the understanding of banana fruit ripening and quality deterioration [4,5]. However, considering the complexity of the ripening processes, a large number of genes and proteins involved in ripening and senescence have not been identified in the banana. 

The recent development of omics technologies, such as transcriptomics, proteomics, and metabolomics, may provide a promising means to reveal the complexity of post-harvest fruit physiology. Transcriptomic and proteomics analysis in the pear [6], apple [7], peach [8], and many other fruits have provided insights into genes and proteins involved in fruit development, ripening, and senescence. In the case of banana, several studies have been reported on the transcriptome and proteome of banana fruit. D’Hont et al. [9] showed that acetylene treatment induces changes in gene expression in banana fruit (pulp and peel), independent of the ability to initiate ripening in response to ethylene by RNA-seq. Asif et al. [10] compared differences in gene expression between ripe and unripe banana pulp tissues and identified major metabolic networks involved in fruit ripening. Toledo et al. [11] compared the proteins of banana pulp at the pre-climacteric and climacteric stages and identified six significantly up-regulated proteins. The above research is beneficial to revealing the mechanism underlying fruit ripening and quality deterioration in banana. However, previous studies were performed using combined samples of pulp and peel [9], or pulp tissue [10], which cannot reflect the changes in the gene expression of banana peel. On the other hand, due to the incomplete banana protein database, only a few of the differentially expressed proteins in the pulp were identified [11]. 

More evidence has shown that the patterns of gene expression signify a sharp difference between the pulp and peel in the banana during ripening. Inaba et al. [12] suggested that ethylene biosynthesis in ripening banana fruit may be negatively controlled in the pulp tissue, but positively in the peel tissue. Elitzur et al. [4] investigated the regulation of MADS-box gene expression during ripening of the banana and found that two independent ripening programs exist in the pulp and peel of banana fruit, which involve the activation of MaMADS2/4/5 in the pulp, but MaMADS1/3 in the peel. Considering the importance of the peel in determining the quality of edible banana, it is necessary to use combined ‘omics’ technologies to develop a comprehensive understanding of ripening-related physiological processes in the peel of banana fruit.

To better understand the mechanism underlying the ripening of harvested banana fruit, we conducted an integrated transcriptomic, proteomic, and metabolomics analysis of banana peel during ripening. The characterization of these identified genes, proteins, and metabolites clearly reflected the dynamic changes in gene and protein expression in banana peel during ripening, hence increasing our knowledge of the complex mechanisms that regulate banana peel ripening.

## 2. Materials and Methods 

### 2.1. Plant Materials and Treatment

Banana (*Musa acuminate* L. AAA group, cv. Brazilian) fruit were harvested at mature green stage (110 days after anthesis) from a commercial orchard in Guangzhou, China. Fruit fingers with a uniform shape, color, and size were selected. In the first experiment, the fruit was stored at 25 °C and 85% to 90% relative humidity, and sampled at 1, 8, 15, 17, 19, and 21 days after harvest. In the second experiment, the fruit were dipped for 10 min in distilled water (control), or 0.1 mM 3-indoleacetic acid (IAA) solution, and then stored at the above-mentioned conditions, and sampled at 1 and 15 days after treatment. Each treatment consisted of three biological replicates with each replicate containing 12 fruit fingers. Peel tissues were taken from the middle part of each finger and ground into a powder in liquid nitrogen and stored at −80 °C for further analysis. 

### 2.2. Determination of Physiological Parameters during Fruit Ripening

Fruit color was determined using a Minolta Chroma Meter CR-400 (Minolta Camera Co. Ltd., Osaka, Japan). Fruit firmness was measured with a penetrometer GY-1 (Hangzhou Scientific Instruments, Hangzhou, Zhejiang, China). Ethylene production rate and IAA content were measured in accordance with the method of Pan et al. [8]. The contents of adenosine triphosphate (ATP), adenosine diphosphate (ADP), and adenosine monophosphate (AMP) were determined by using high performance liquid chromatography (HPLC) as described by Wang et al. [13]. The respiration rate was determined according to the method of Wang et al. [13]. Starch content was determined according to the method of Gao et al. [14]. The contents of cell wall components, including cellulose, hemicellulose, water-soluble pectin, acid-soluble pectin, chelaing-soluble pectin, and lignin, were determined according to the method of Zhao et al. [15].

### 2.3. RNA-Seq (Quantification) Analysis 

Total RNA was extracted from banana peel using a Qiagen RNeasy Kit (Qiagen, Duesseldorf, Germany), according to the manufacturer’s instructions. After the RNA integrity assessment using the Bioanalyzer 2100 system (Agilent Technologies, Santa Clara, CA, USA), messenger RNA (mRNA) was purified using poly-T oligo-attached magnetic beads and then fragmented. First strand cDNA was synthesized using random oligonucleotides and SuperScript II. Second strand complementary DNA (cDNA) synthesis was subsequently performed using DNA polymerase I and ribonuclease H. After adenylation of the 3’ ends of DNA fragments, Illumina PE adapter oligonucleotides were ligated to prepare for hybridization. The library fragments were purified with an AMPure XP system (Beckman Coulter, Beverly, MA, USA). The DNA fragments with ligated adaptor molecules on both ends were selectively enriched using Illumina PCR Primer Cocktail (Illumina Inc., San Diego, CA, USA) in a 10-cycle polymerase chain reaction (PCR). After cluster generation, the library preparations were sequenced on an Illumina Hiseq 2000 platform (Illumina Inc). Single-read sequencing (expected library size: 200 base pairs; read length: 50 nucleotides) was performed in the present study.

Clean reads were obtained by removing reads containing adapters, reads containing ploy-N, and low-quality reads from raw data. The uniquely mapped reads for a specific transcript were counted by mapping reads to banana genome sequences (http://banana-genome.cirad.fr/) using short oligonucleotide alignment program (SOAP, version 2) [16]. Then, the reads per kilobase million (RPKM) value for each transcript was measured in reads per kilobase of transcript sequence per million mapped reads [17]. The false discovery rate method was used to determine the threshold of the *p* value in multiple tests. False discovery rate (FDR) ≤ 0.001 and an absolute value of log2Ratio ≥ 1 was the threshold to judge the significance of differentially expressed genes.

### 2.4. Quantitative Real-Time Polymerase Chain Reaction Analysis

The cDNA was synthesized by using a PrimeScript™ RT reagent Kit with gDNA Eraser (Takara, Otsu, Japan). Gene specific primer pairs (Appendix A) were designed with the Primer Express software (Applied Biosystems, Foster City, CA, USA). Actin was selected as the reference gene. Quantitative reverse transcription-PCR was performed as described by Yun et al. [18]. Three biological replicates were conducted. Output data were generated by Sequence Detector version 1.3.1 software (PE Applied Biosystems) and then statistically analyzed using the 2^−ΔΔCT^ method.

### 2.5. Two-Dimensional Gel Electrophoresis (2-DE) and Protein Identification

Peel tissues (20 g) was extracted with 30 mL of 10% trichloroacetic acid in acetone. Then, total protein extraction was conducted by the method of Yun et al. [19]. Protein concentration was measured with an RCDC^TM^ protein assay kit (Bio-Rad, Hercules, CA, USA) using BSA as the standard. Isoelectric focusing (IEF) was done essentially as described by Yun et al. [19], but with a slight modification. A 2 mg sample of protein was used to passively rehydrate the 17 cm pH 4 to 7 immobilized pH gradient (IPG) strip overnight. Isoelectric focusing was done in a Protean IEF Cell (Bio-Rad), and 80,000 Vh was used in the focusing step. After IEF, the strips were equilibrated twice for 15 min [18]. Second-dimensional sodium dodecyl sulfate/polyacrylamide gel electrophoresis (SDS-PAGE) was done in 12% polyacrylamide gels at 80 V for 0.5 h and then at 180 V for 5.5 h. More than four replicate gels were run for each sample. Proteins in the electrophoresis gels were visualized by staining with coomassie brilliant blue R-250 [19]. The gels were scanned with a GS-710 densitometer (Bio-Rad). Spot detection, quantification, and matching were done for each set of gel with PDQuest 2-D analysis software version 8.0 (Bio-Rad). Data were normalized as a percentage of the total density of all spots on the corresponding gel and then analyzed by 1.5-fold comparative analysis and one-way analysis of variance [18]. Experimental molecular mass and isoelectric point (pI) were calculated from digitized 2-D electrophoresis images using standard molecular mass marker proteins and IPG strip pH 4 to 7.

Protein spots were excised manually, and then de-stained overnight with 200 μL of 50 mM ammonium bicarbonate in 40% ethanol. After being incubated twice with acetonitrile, gels were dried and digested using trypsin solution (75 ng/μL in 50 mM ammonium bicarbonate). Peptide solution was spotted onto a stainless steel target plate. Tryptic peptides were analyzed by matrix-assisted laser desorption/ionization time-of-flight tandem mass spectrometry (MALDI-TOF MS/MS) with an Applied Biosystems mass spectrometer (model 4800, Framingham, MA, USA) to acquire MALDI and MS/MS spectra [20]. Resulting peak lists were searched against the banana protein database (http://banana-genome.cirad.fr/) using MASCOT v2.1.03 software (Matrix Science, London, UK). The search was performed using the following settings: Plants trypsin, one missed cleavage, fixed modifications of carbamidomethyl, variable modifications of oxidation, peptide tolerance 100 ppm (parts per million), fragment mass tolerance ±0.5 Da, and peptide charge 1+. Only peptides with MS/MS ion scores significantly (*p* < 0.05) exceeding the MASCOT identity or extensive homology threshold were reported. 

### 2.6. Metabolic Profiling

Samples collected at 1, 15, 19, and 21 days after harvest were used for the differential metabolic profiling analysis. Primary metabolites were detected using gas chromatography coupled to mass spectrometry (GC-MS) as described by Yun et al. [21]. Briefly, peel sample (300 mg) was extracted with 2700 µL of methanol. 300 µL of ribitolin (0.2 mg mL^−1^) was added as a quantification internal standard. Extracts were incubated in 50 µL of methoxyamine hydrochloride in pyridine (20 mg mL^−1^) for 30 min at 50 °C and then treated using 50 µL of BSTFA (N,O-bis(trimethylsilyl)trifluoroacetamide) at 60 °C for 40 min. One µL of sample was injected into the gas chromatograph system through a fused-silica capillary column DB-5 MS stationary phase. The program of GC-MS was as follows: Injector temperature was 250 °C; carrier gas flow rate was 1.0 mL min^−1^; column temperature was held at 100 °C for 1 min, then increased to 184 °C at a rate of 3 °C min^−1^, increased to 190 °C at a rate of 0.5 °C min^−1^, increased to 280 °C at 15 °C min^−1^; flow rate of the carrier helium (99.999%) gas was 1 mL min^−1^. Mass spectrometry operating parameters were as follows: Ionization voltage at 70 eV, ion source temperature at 200 °C, interface temperature at 250 °C. Total ion current spectra were recorded in the mass range of 45 to 600 atomic mass units. Volatile compounds were collected and detected according to the method of Jing et al. [22]. Briefly, 4 g of peel was ground and homogenized with 4 mL of sodium chloride saturated solution. After being held at 40 °C for 15 min, volatile compounds were collected for 45 min using headspace solid-phase microextraction (HS-SPME). The GC-MS analysis was performed using a GC-2010 gas chromatography (GC-MS-QP2010 Plus, Shimadzu Corporation, Kyoto, Japan) equipped with the DB-5 MS stationary phase fused-silica capillary column (30 m × 0.25 mm i.d., 0.25 μm, Agilent Technologies Inc., Santa Clara, CA, USA). The total ion current spectra scanned range from 30 to 550 m/z. The mass spectra were compared with compounds from the NIST05 and NIST database. After the normalization analysis according to the total peak area, the relative quantification of these compounds was calculated according to the peak area ratios of the quantitation ions of the internal standard (cyclohexanone).

### 2.7. Bioinformatics Analysis

Gene ontology (GO) analysis was performed by mapping the differentially expressed genes or proteins with gene ontology terms (http://en.wikipedia.org/wiki/Gene-Ontology). The basic local alignment search tool (BLAST) based on sequence similarity was performed according to *Arabidopsis* protein release TAIR10 (ftp://ftp.arabidopsis.org/home/tair/Sequences/blast_datasets/TAIR10_blastsets/) using the default settings. The metabolic clustering analysis was performed using MapMan software (3.6.0RC1, Aachen, Germany). Ontology-based clustering algorithm (CLUGO) analysis was performed by using CLUGO plug-in version 2.2.4 of Cytoscape version 3.3.0. The protein–protein interactions network was constructed using STRING version 10.5, and the result was presented using Cytoscape version 3.3.0.

### 2.8. Statistical Analysis

Data presented in this study are the mean values of three biological replicates. Data are expressed as the mean ± standard deviation. The significance of difference among different samples was calculated using statistics package for the Social Sciences version 7.5 (SPSS, Inc., Chicago, IL, USA). Principal component analysis in R was performed in volatile compounds analysis. The analysis of two-way orthogonal partial least squares (O2PLS) was performed using SIMCA version 15.0 (MKS Umetrics AB, Umeå, Sweden). 

## 3. Results

### 3.1. Ripening and Physiological Characteristics of Harvested Banana Fruit

Turning yellow is the most obvious change and the major criterion used by consumers to determine whether banana fruit are ripe or not. As shown in Figure 1A,B, there was a slight change in peel color from 1 d to 15 days, while the hue angle value decreased significantly after 17 days of storage, indicating that the degradation of chlorophyll and the synthesis of carotenoid occurred. The whole process can be divided into four stages based on fruit color change, including the mature green stage (0 to 8 days after harvest), green > yellow stage (8 to 17 days after harvest), green < yellow stage (19 days after harvest), and whole yellow stage (21 days after harvest). Softening is another important characteristic of banana fruit ripening, which is the major factor limiting the shelf life of banana fruit. In the present study, fruit firmness decreased from the initial value of 49.67 N cm^−2^ to 4.17 N cm^−2^ after 21 days of storage at 25 °C (Figure 1B). According to the firmness characteristic, the softening process was obviously divided into four stages: Hard maturation (49.67 N cm^−2^), a slight decrease in firmness (31.00 to 35.00 N cm^−2^), a moderate decrease in firmness (14.50 N cm^−2^), and a severe decrease in firmness (4.17 N cm^−2^). Based on the overall changes in the fruit color and firmness, the samples from four ripening stages, i.e., 1, 15, 19, and 21 days after harvest, were selected for the transcriptomic, proteomic, and metabolomics analysis, which were defined as P1, P2, P3, and P4, respectively.

Ethylene is a hormone that promotes ripening and senescence in fruits. As shown in Figure 1C, banana fruit ripening was accompanied by a sharp increase in the ethylene production rate, followed by a rapid increase in the respiration rate. Similarly, the content of IAA, another ripening-related hormone, showed a rapid increase in P2, and subsequently decreased gradually (Figure 1C). 

Fruit softening is related to changes in turgor, and degradation of cell wall components. Starch is the bulk polysaccharide present in banana fruit and its hydrolysis also leads to loosening of the cell structure [23]. As shown in Figure 1D, the contents of total starch and amylose rapidly decreased in P3 and P4. The cell wall components, including acid-soluble pectin, chelating-soluble pectin hemicellulose, and lignin, rapidly degraded in P3 and P4 (Figure 1D).

The contents of ATP, ADP, and level of energy charge generally decreased while AMP content increased as storage time progressed. There were low levels of ATP, ADP, and energy charge, but a relatively high level of AMP in the banana peel in P4 (Figure 1E).

### 3.2. Transcriptomic Analysis of Banana Fruit during Ripening

To study the gene expression of banana peel during ripening by transcriptomics, cDNA libraries from four ripening stages were subjected to Illumina sequencing. More than seven million reads were detected in each sample, in which clean reads accounted for more than 92%, and perfectly matched reads was more than 2,641,180 in each replicate. The distribution of gene coverage showed that reads (90–100%) were more than 23% and reads (80–90%) were more than 11% in each replicate. 

A total of 5784 genes were differentially expressed in the peel during banana fruit ripening. Compared with P1, 858, 1516, and 824 genes were up-regulated while 1001, 3382, and 2906 genes were down-regulated in P2, P3, and P4, respectively (Figure 2A). Detailed information on the differentially expressed genes is shown in Appendix A. A total of 1254 genes showed transcript abundance differences in all P2, P3, and P4, and 1917 genes in both P3 and P4 (Figure 2B). Specifically, 1501 genes were only found in P3 (Figure 2B). As the larger number of differentially expressed genes, it is suggested that P3 was a critical period during banana ripening.

Gene ontology categories were assigned to evaluate the potential functions of genes with significant transcriptional differences in the peel of banana fruit during ripening using the Blast2GO program (http://www.blast2go.org/start_blast2go#download_blast2go). In terms of biological processes, 23 categories were classified. The biggest one was related to the cellular process (2127 genes), followed by the metabolic process (2049), single-organism process (1313), response to stimulus (1121), biological regulation (723), and localization (617) (Figure 2C). The significant enrichment categories according to molecular function were catalytic activity (1673), binding (1557), transporter activity (223), structural molecule activity (174), nucleic acid binding transcription factor activity (71), electron carrier activity (37), antioxidant activity (33), molecular transducer activity (30), enzyme regulator activity (22), protein binding transcription factor activity (14), nutrient reservoir activity (5), and receptor activity (1) (Figure 2C).

To confirm the reliability of RNA-seq data, the expressions of 18 genes, randomly selected from significantly up or down-regulated genes, were analyzed using qRT-PCR. These genes were related to signal transduction, transcription regulation, photosynthesis, cell wall metabolism, energy metabolism, and other metabolisms. The results showed that the expression patterns of all 18 genes were consistent between qRT-PCR and RNA-seq analyses (Figure 3).

### 3.3. Proteomic Analysis of Banana Fruit during Ripening

To exploit ripening-related proteins, 2-DE coupled MALDI-TOF MS/MS were applied in proteomic profiling. After IEF, PAGE electrophoresis, and coomassie brilliant blue (CBB) staining, more than 700 reproducible protein spots in all replicates, with the pI and molecular mass ranging from 4.0 to 7.0 and 10.0 to 100.0 kDa, respectively (Appendix A), were detected by PDQuest 2-D analysis software. Of these protein spots, 108 spots exhibited significantly differential expression in the peel of banana fruit during ripening. Finally, 94 spots were successfully identified using the banana genome (http://banana-genome.cirad.fr/) as the reference sequence. Compared with P1, 31, 39, and 42 proteins were up-regulated while 29, 37, and 47 proteins were down-regulated in P2, P3, and P4, respectively (Figure 4A). Moreover, 52 proteins were differentially expressed in all P2, P3, and P4, compared with P1 (Figure 4B). The specific locations of proteins on 2-DE gels are shown in the gels in Appendix A and detailed information on these proteins is provided in Appendix A. According to gene ontology annotations and the literature, the identified proteins could be classified functionally into 11 categories (Figure 4C). A considerable enriched cluster was the oxidation-reduction process (16 proteins), followed by protein degradation and modification (13), carbohydrate metabolic process (12), other metabolic process (11), photosynthesis (11), cell wall organization (6), secondary metabolic process (6), signal (6), fruit ripening (5), energy metabolic process (5), and aging (3) (Figure 4C). 

### 3.4. Metabolomics Profiling of Banana Fruit during Ripening

A total of 63 primary metabolites were identified in the banana peel by GC-MS analysis, mainly including sugars, organic acids, amino acids, alcohols, fatty acids, and other metabolites, of which 36 metabolites were up- or down-regulated significantly in P2, P3, and P4, compared with P1 (Figure 5). Interestingly, the contents of all 13 kinds of sugars were increased in P4 compared with P1, with more than a 100-fold increase in the contents of D-fructose 1 (104.01-fold), D-glucose (148.04), D-mannopyranose (186.83), D-fructose 2 (125.63), and D-tatatofuranose (444.87). Seven of them were increased more than 3-fold in P2, P3, and P4 compared with P1, respectively. These results suggest that there might be a sharp hydrolysis from starch to small molecular sugars, especially at the P4 stage. The contents of all seven organic acids were increased in P2 and P4 compared with P1, while the contents of 2-ketoglutaric acid, butanedioic acid, and butanedioic acid decreased in P3 (Figure 5). Only the contents of valine and lysine were increased slightly during banana ripening; other amino acids showed no significant difference (Figure 5).

A total of 93 volatile compounds were identified in the banana peel, 81 of which significantly accumulated during ripening, especially in P4, including esters, alcohols, acids, hydrocarbons, ketones, volatile phenols, ethers, anhydrides, and aldehydes. Principal component analysis (PCA) showed that the first and second principal components accounted for 97.11% of the variance in the data set (Figure 6A), and the sample separation was highest in the first principal component (PC 1). During ripening, the difference was mainly observed in P4 (Figure 6A). The other groups were not obviously separated. These results indicated that the metabolic changes mainly occurred at 21 days. Of the volatile compounds, esters accounted for the major changes in volatile aroma synthesis (Figure 6B). Sixteen esters increased at least 100-fold at P4 compared with at P1, including n-butyl butanoate, 2-methyl isoamyl butyrate, 1-methylbutyl butanoate, 2-tridecanyl valerate, vinyl butyrate, hexyl isovalerate, isobutyl butyrate, (2Z)-2-pentenyl butyrate, 2-pentanyl valerate, 3-tridecanyl valerate, isoamyl isobutyrate, 1,2-ethanediyl dibutanoate, 2-acetoxyoctane, hexyl butyrate, isobutyl isovalerate, and isopentyl hexanoate (Appendix A).

### 3.5. Correlation Analysis of Transcriptomic and Proteomic Data 

An analysis of the correlation between 94 differentially expressed proteins from proteomic data and their corresponding mRNAs from transcriptomic data was performed. Expression of differentially expressed proteins correlated well with those of their corresponding genes for P3/P1 and P4/P1, with a *p* value of less than 0.01 and an R value of 0.45 in P4/P1 (Figure 7). For P2/P1, the *p* value and the R value were 0.256 and 0.118, respectively (Figure 7). It seems that P3 might be a crucial stage during banana peel ripening while P2 might be a stage at which the protein expression could not immediately respond to gene expression. The total numbers of the correlative element (up- and down-regulated) between proteins and genes were 15, 25, and 43 in P2/P1, P3/P1, and P4/P1, respectively (Figure 7), exhibiting a trend of an increase, especially for down-regulated elements (7, 15, and 31 in P2/P1, P3/P1, and P4/P1, respectively) (Figure 7). Interestingly, the expression trends of four up-regulated proteins (P13, 79, 85, and 91 in Appendix A) and five down-regulated proteins (P2, 29, 46, 48, and 53 in Appendix A) remained consistent in P2, P3, and P4 compared with P1. The down-regulated proteins were mainly involved in photosynthesis while the up-regulated proteins were mainly related to cell wall organization and signal transduction. 

To further investigate the role of the differentially expressed genes or proteins in banana peel ripening, 1988 up-regulated genes or proteins were used for protein–protein interaction network construction. The results revealed some key proteins, including acetyl-coenzyme A carboxylase, receptor protein kinase (transmembrane kinases 1), inactive leucine-rich repeat receptor-like protein kinase, glutamate dehydrogenase, ATP-citrate synthase, glycine dehydrogenase, and glucose-6-phosphate isomerase (Appendix A), which were mainly involved in signaling, glycolysis, tricarboxylic acid cycle, amino acid metabolism, energy metabolism, polysaccharide degradation, and protein modification (Appendix A). According to the transcriptomic data, the genes involved in carbohydrate metabolism, secondary metabolism, energy metabolism, and proteins modification are also important for banana fruit ripening.

To explore the correlation between differentially expressed genes/proteins and metabolites, O2PLS was performed using SIMCA version 15.0 (Umetrics). O2PLS analysis indicated that the explained variation in the X matrix (R2X[1] and R2X[2]) accounted for 0.784 of the variance in the correlation matrix (Appendix A). The whole ripening process of the banana peel can be divided into three parts: P1, P2/P3, and P4. The contents of soluble sugar and volatile compounds significantly increased, while a great number genes and proteins related to cell wall degradation, volatile aroma synthesis, and protein modification were specifically up-regulated in P4. Curiously, a large number of genes and proteins related to signaling and the metabolic process were significantly up-regulated, but few metabolites were significantly increased in P2 and P3.

### 3.6. Specifically Expressed Signal Transduction-Related Genes and Proteins during Banana Ripening

A total of 268 genes and five proteins related to signal transduction were differentially expressed during banana ripening, 100 of which were up-regulated. CLUGO analysis showed that regulation of the ethylene-mediated signaling pathway was the main cluster (Appendix A). In addition, auxin homeostasis was also clustered in the CLUGO analysis (Appendix A). 

In the present study, 10 genes related to the auxin signal were up-regulated during banana ripening. To investigate the effect of auxin on banana ripening, mature banana fruit were treated with 0.1 mM IAA for 10 min. The IAA treatment accelerated fruit ripening, accompanied by earlier yellowing and softening (Figure 8A). We also analyzed the expression of 16 ripening-related genes using qRT-PCR and found that most genes were up-regulated in IAA-treated fruit, including auxin signaling-related genes, ethylene signaling-related genes, and cell wall degradation-related genes (Figure 8B). These results indicated that exogenous IAA treatment accelerated the ripening of harvested banana fruit.

### 3.7. Specifically Expressed Transcription Factor Genes and Proteins during Banana Ripening

In the present study, 955 transcription factor genes were detected, 158 of which were differentially expressed. After CLUGO analysis, 60 up-regulated transcription factor genes were clustered in the ethylene-mediated signaling pathway (Appendix A), while 98 down-regulated transcription factor genes were related to the gibberellin stimulus and the regulation of photomorphogenesis (Appendix A). Specifically, 18 ERFs (ethylene response factor), 6 bHLHs (basic helix-loop-helix), 5 WRKYs, 3 myeloblastosis oncogens (MYBs), and 2 myelocytomatosis viral oncogenes (MYCs) were up-regulated at the transcription level (Appendix A). In addition, MYB-related protein, Pp2, was up-regulated at the proteomic level (Appendix A).

### 3.8. Specifically Expressed Metabolic Process-Related Genes and Proteins during Banana Ripening

All the identified metabolic process-related genes and proteins were used to perform metabolic clustering analysis using MapMan software (3.6.0RC1, Aachen, Germany). A series of metabolic processes associated with light reactions, photorespiration, tetrapyrrole, and amino acids were down-regulated (Appendix A) while the up-regulated processes were related to cell wall and lipid metabolism, carbohydrate and energy metabolism, and secondary metabolism (Appendix A). 

Thirty-five genes and six proteins related to cell wall metabolism were up-regulated, mainly including ten xyloglucan endotransglucosylase/hydrolase (XTHs), two pectate lyases, three polygalacturonases, two endoglucanases, and two pectinesterases (Appendix A), which were consistent with the degradation of cell wall components (Figure 1C). 

Thirteen starch and sucrose metabolism-associated genes and proteins were up-regulated, of which seven were related to starch degradation, and six were responsible for sucrose degradation (Figure 9). Up-regulation of starch degradation-related genes were consistent with the decrease of the total starch content in banana fruit peel (Figure 1C). A great number of genes and proteins related to energy metabolism showed up-regulated expression during fruit ripening, including eight glycolysis-related genes, four tricarboxylic acid cycle enzymes, five alcohol dehydrogenases, two ATP-citrate synthases, two cytochrome c oxidases, and three ADP, ATP carrier proteins (Appendix A). 

Twenty-two flavonoids synthesis-related genes and 42 phenylpropanoid metabolism-related genes were up-regulated during banana fruit ripening. In addition, the up-regulation of 14 volatile compounds metabolism-related genes was also observed at P4, which might play an important role in the synthesis of the volatile aroma (Appendix A).

### 3.9. Specific Genes and Proteins Related to Protein Modification and Degradation during Banana Ripening

A total of 66 genes and four proteins related to protein modification were up-regulated during banana ripening, while 78 genes and five proteins were down-regulated (Appendix A). Specifically, 10 heat shock proteins, especially large molecular weights of heat shock proteins, were up-regulated at the mRNA level. The down-regulated genes and proteins were mainly related to chlorophyll a-b binding. In protein degradation, 19 genes and two proteins were up-regulated during banana ripening (Appendix A), which were mainly associated with ubiquitin-dependent protein degradation. 

## 4. Discussion

Banana is a typical climacteric fruit, which undergoes a series of highly coordinately physiological and biochemical processes after harvest to achieve the best edible quality, such as respiratory climacteric, pulp softening, peel de-greening, and volatile aroma production [1]. Banana peel and pulp have very different ripening attributes, as well as, possibly, the regulatory mechanism of ripening [4,12]. The present work performed a study of integrated transcriptomic, proteomic, and metabolomics analysis of the banana peel during ripening. Herein, we mainly discuss the molecular mechanisms underlying banana peel ripening in terms of hormone signaling, transcription regulation, cell wall degradation, volatile aroma synthesis, protein modification, and energy metabolism. 

### 4.1. Hormone Signaling 

Plant hormones play significant roles in regulating fruit development and ripening. Various receptors and key signaling components of these hormones were identified. In the present study, a total of 268 genes and six proteins associated with hormone signaling were differentially expressed during banana peel ripening, of which 100 genes were up-regulated, especially in P2 and P3. The majority of differentially expressed signaling-associated genes were related to ethylene and the auxin mediated signaling pathway (Appendix A). 

Ethylene plays a crucial role in banana fruit ripening [24]. The sharp increase in ethylene preceded fruit softening and turning yellow, indicating that ethylene initiated banana fruit ripening (Figure 1B,C). Currently, ethylene perception and signal transduction has been well characterized in the model plant species, *Arabidopsis* [25]. However, fruit have a specific regulatory mechanism of ethylene signaling, especially during fruit ripening [26]. In the present study, the expression of two ethylene receptor genes, as the ethylene signal direct transmitter, were up-regulated at the early stage of ripening (i.e., P2 and P3). Ethylene signaling is transmitted to ETHYLENE INSENSITIVE (EIN) 2 through constitutive triple response 1 (CTR1) in the cytoplasm, then amplified by a transcription factor cascade, including EIN3, ERF, etc., within the nucleus [27]. The level of EIN3 protein is also regulated by ubiquitination and proteasome degradation via EBF1 and EBF2 [28]. Surprisingly, only one EIN3 gene and three EBF1 genes were observed as being up-regulated at the P2 stage, while four EIN3 genes were down-regulated at the P3 and P4 stages, accompanied by the down-regulated expression of two EBF genes. Therefore, ethylene signaling was complex during banana ripening. As mentioned above, a number of ERF genes, downstream of the ethylene signaling, were up-regulated during banana fruit ripening (Appendix A). Taken together, some genes involved in ethylene receptors, ethylene signaling regulator, and ethylene response transcript factors were up-regulated, indicating that the ethylene signal played an important role during banana ripening. 

Various hormone signals have synergistic effects with ethylene signaling, especially during fruit ripening [3,26]. It is well documented that auxin is involved in cell expansion and division, tissue differentiation, organ development, and a range of physiological processes [29]. Recently, increasing evidence shows that auxin also plays an important role in the regulation of fruit ripening [8,26]. In the present study, a larger number of auxin signaling-related genes were up-regulated ((Appendix A) and the IAA content significantly increased (Figure 1C) during banana ripening, indicating that auxin likely plays an important role in the regulation of the ripening process of harvested banana. These differentially expressed genes were involved in auxin homeostasis, perception, signaling, and regulation of transcriptional responses (Appendix A). For the auxin signaling pathway, Aux/IAAs (AUXIN/INDOLE-3-ACETIC ACID) and ARFs (auxin response factor) are the major molecular components [29]. In this study, one ARF was up-regulated while two repressors of auxin-responsive transcription (*Aux/IAAs*) were down-regulated in P2 compared to P1, implying that auxin signaling might be activated during banana ripening (Appendix A). Furthermore, the application of exogenous IAA accelerated banana fruit ripening (Figure 8A). Auxin has also been reported to accelerate fruit ripening in the peach [8] and plum [30]. It is suggested that auxin signaling might play a positive role in banana ripening. Although many ARFs and Aux/IAAs were also down-regulated in P3 and P4 compared to P1, which might be caused by the difference of the banana ripening stages, as P2 might be a signaling stage and P3 might be a physical changing stage. Our results also showed that both auxin signaling and ethylene signaling-related genes were up-regulated after IAA treatment. The expression of some ethylene biosynthesis-related genes (*ACO* and *ACS*) and softening-related genes (*XTH* and *PG*) were significantly up-regulated by exogenous IAA treatment, suggesting that IAA treatment induced ethylene biosynthesis and the ripening-related process. D’Hont et al. [9] showed that acetylene, an ethylene analogue, up-regulates the expression of a great number of auxin signaling-related genes and activates the ripening process of harvested banana fruit. Busatto et al. [31] found that ethylene/auxin cross-talk probably exists in apple, which is regulated by a transcription circuit stimulated by inference at the ethylene receptor level. It seems that the cross-talk between ethylene signaling and auxin signaling played an important role in the peel ripening of harvested banana fruit in the present study. However, the interaction between ethylene and auxin signals is poorly understood and further analysis is needed to identify the interacting nodal genes.

### 4.2. Transcription Factors

Transcriptional regulation plays a vital role in fruit development and ripening as well as abiotic stresses. In the present study, 955 transcription factors (TFs) were detected, of which 127 TFs were significantly differentially expressed across the four ripening stages. Among the differentially expressed TFs, ERF family members were the most abundant TFs. The ERF transcription factor super family is one of the important regulators in plants, which are involved in various physiological processes, including plant growth and development as well as in response to hormones, and biotic and abiotic stresses [32]. Li et al. [33] found that MdERF2 negatively affects ethylene biosynthesis in the apple during fruit ripening by suppressing MdACS1 transcription. Han et al. [34] reported that the MaERF1 transcription factor recruits histone deacetylase MaHDA1 and represses the expression of MaACO1 and expansins during banana fruit ripening. Kuang et al. [35] showed that MaDREB2 may serve as both a transcriptional activator and repressor to participate in the regulation of ripening-related genes in harvested banana fruit ripening. In the present study, most of the differentially expressed *ERFs* were up-regulated. Considering the multigene-encoded TFs, the down-regulated expression indicated ERFs play differential roles during various stages of ripening and fruit development. Further, it is worth noting that the surprisingly higher up-regulation of several *ERF* genes occurred in the P2 stage compared with the P3 and P4 stages, which confirmed that ERF participated in the initiation of banana fruit ripening.

Basic helix-loop-helix transcription factors were another large group of TFs with differential expression during banana fruit ripening. Basic helix-loop-helix TFs have been implicated in the regulation of a wide range of processes, including plant development, secondary metabolism, hormone signaling, and response to biotic and abiotic stresses alone or in combination with other TFs, such as MYBs. It was reported that bHLH participates in the regulation of carotenoid and anthocyanin biosynthesis in citrus [36] and sweet cherry [37], respectively. In addition, Peng et al. [38] proposed that bHLHs were involved in the response of banana fruit to low temperature stress in the regulatory network of the C-repeat binding factor (CBF)-inducer of CBF expression (ICE) cold signaling pathway. Banana fruit ripening is accompanied by the synthesis of carotenoid. It is suggested that bHLH TFs might play an important role in regulating the biosynthesis of carotenoid in peel during banana fruit ripening.

Apart from the *ERF* and *bHLH* genes, the MYB and WRKY families were highly differentially expressed during banana fruit ripening. Our results were inconsistent with those reported by Asif et al. [10], who compared the transcriptome profiling of ripe and unripe pulp tissue of banana fruit and found that the most abundant differentially expressed TFs were MADS and MYB-related gene families. ERF, bHLH, MADS, and MYB TFs are encoded by multigene families in the banana [9]. The inconsistency indicates that the expression of some members of these TFs family is tissue-specific. Further, it was found that more TFs were up-regulated in P2 than in P3 or P4 (Appendix A), but the larger number of up- or down-regulated genes was shown in P3 (Appendix A). We speculated that P2 might be a signaling stage, at which a larger number of signaling-related genes are up-regulated, while P3 might be a physical changing stage for banana fruit ripening, at which a larger number of functional genes are up-regulated.

### 4.3. Cell Wall Degradation

Plant cell walls mainly consist of complex networks of polysaccharides (cellulose, hemicelluloses, and pectins), proteins, and lignin. Depolymerization of pectin and hemicellulose plays an important role in fruit ripening, leading to the disassembly of the cellulose and hemicellulose network and the decrease in fruit firmness [39]). The involvement of polygalacturonase (PG), pectin esterase (PE), and pectate lyase (PL) in the enzymatic degradation of pectin polysaccharides has been well documented [40]. PG hydrolyzes α-1,4-glycosidic bonds between galacturonic acid residues, following de-esterification of pectin by PE. In the present study, the expression of one *PG* gene (GSMUA_Achr2T14330_001) and one *PE* gene (GSMUA_Achr7T11480_001) were enormously up-regulated as fruit ripened, while two other *PG* genes and two other *PE* genes were significantly up-regulated only during late ripening stages, indicating that different *PG* or *PE* genes worked at different softening stages. In addition, two genes encoding pectinacetylesterase (PAE), functioning as the de-esterification of pectin, were significantly up-regulated throughout ripening or during late ripening, which was a necessary and beneficial complement to PE. These genes showed co-expression patterns during banana fruit ripening, which further confirmed the collaboration of PG and PE/PAE in the disassembly of pectin polysaccharides in banana peel that are required for fruit softening. PL, known as pectate transeliminases, catalyze the eliminative cleavage of de-esterified pectin [41]. However, it is thought that PLs are secreted mainly by plant pathogens, resulting in the maceration of plant tissues [42]. Interestingly, the abundance of PL-like sequences in banana genomes (24 genes in the banana genome encoding PL-like proteins) [9] strongly suggests an important role of these enzymes in various banana ripening stages. In the present study, *PL15* and *PL22* were up-regulated throughout the softening process, suggesting that PL might be implicated in the degradation of pectin in banana peel.

Several genetic studies have shown that the degradation of pectins is not sufficient to cause fruit softening [2] and the degradation of other cell wall components is also required for fruit softening [40]. In land plants, xyloglucans constitute most hemicelluloses’ and celluloses’ polysaccharides. The modification of hemicelluloses and celluloses is involved in the degradation of the cellulose xylogluc/pectin network, which contributes to fruit softening [43]. The xyloglucan endotransglucosylase/hydrolase (XTHs) family is believed to play a key role in this process, which catalyzes the nonhydrolytic cleavage and rearrangement of xyloglucan chains through xyloglucan endotransglucosylase activity or directly through the hydrolysis of xyloglucan through xyloglucan endohydrolase activity [44]. In the present study, XTH family members were the most abundant among all differentially expressed genes associated with the cell wall modification, of which 10 *XTH* genes were up-regulated, which is consistent with the significant decrease of cellulose and hemicellulose contents (Figure 1D). Our previous study demonstrated that xyloglucan was the main hemicellulose polysaccharide in banana fruit and underwent significant disassembly during fruit softening. These results in this study further confirmed that the degradation of hemicelluloses by XTH play a critical role in banana fruit softening. 

### 4.4. Synthesis of Volatile Compounds

Volatile metabolites attract pollinators and seed dispersals, stimulate or suppress signaling cascades, and protect against harsh environmental conditions, herbivores, or pathogens, and also contribute to the aroma and flavor of fruits [45]. The volatile aroma is one of the most important factors determining the quality of banana fruit. During banana fruit ripening, volatile compounds were mainly synthesized at P4 while the volatile compounds detected were less in variety and extremely low in content in the peel of banana fruit at other stages of ripening (Appendix A). Esters accounted for 54% of the total volatile aroma content in banana fruit peel (Figure 6B). The most likely precursors for the esters are amino acids and lipids. Arogenate dehydratase, an enzyme participating in phenylalanine, tyrosine, and tryptophan biosynthesis [46], was up-regulated 4-fold at P4, which benefited the synthesis of precursors and volatile esters. Alcohols are another major type of the volatile aroma in banana fruit, which accounted for about 35% of the total volatile aroma content (Figure 6B). The significantly up-regulated alcohol dehydrogenases and aldehyde dehydrogenases (Appendix A) contributed to the synthesis of volatile alcohols [47]. Moreover, the TCA cycle and anaerobic respiration were accelerated during banana ripening, which might provide carbon skeletons and energy for the volatile aroma synthesis [48]. In addition, several basic region/leucine zipper motif (bZIP) family transcription factors and NAC (NAM, ATAF1, 2 and CUC2) domain-containing protein 2-like were significantly up-regulated at P4 (Appendix A), which were also reported to play a vital role in regulating volatile aroma production in ripening kiwifruit [49]. This indicated that bZIPs and NACs might play an important regulatory role in the synthesis of volatile compounds in banana peel. However, the transcription factors that regulate the synthesis and accumulation of aromatic esters have not been confirmed, and further analysis is needed to verify their gene functions.

### 4.5. Protein Modification

Fruit ripening involves protein modification, including biosynthesis, degradation, and folding. In this study, the differentially expressed genes or proteins mainly included heat shock protein (Hsp), proteasome, ubiqutin tagging system, protein secretion, protein disulfide isomerase, and chlorophyll a-b binding. 

Heat shock proteinsare a stress-responsive family of proteins, which play important roles in posttranslational modifications and protein folding/aggregation/disaggregation. Heat shock proteins are known to be involved in abiotic tolerance in plants. There are also reports that fruit ripening provokes the accumulation of Hsps [6]. In this study, Hsp family members were the most abundant (Appendix A). Interestingly, 10 large molecular weight Hsps, especially Hsc70, were up-regulated at the mRNA level, while 3 small Hsps, with a low molecular weight ranging between 15 and 42 kDa, were down-regulated at the protein level. Heat shock congnate 70 kDa protein (Hsc70), a member of the Hsp70 family, is constitutively expressed and involved in protein folding and translocation, which protects proteins from denaturation and dysfunction [50]. The up-regulated expression of Hsc70 genes is required to refold damaged proteins and to carry irreversibly denatured proteins to the proteasome. Small Hsps are mainly reported to function as chaperones and assist in maintaining the fluidity and integrity of the cell membrane [51]. Zou et al. [52] reported that over-expression of *OsHsp23.7* and *OsHsp17.0* in rice resulted in a significant decrease in electrolyte leakage. The decreased expression of small Hsps was unfavorable for the stability of the protein and membrane in a banana peel.

The ubiqutin-protesome system is the important mechanism of protein degradation. Ubiquitin are covalently attached to a diverse array of target proteins through a cascade of reaction catalyzed by three kinds of enzymes: E1-ubiquitin-activating enzyme, E2-ubiquitin-conjugating enzyme, and E3-ubiquitin-protein ligase enzyme [53]. Consequently, the ubiquitin-tagged target proteins are degraded by proteasomes. There are some reports regarding the involvement of the ubiqutin-protesome system in the regulation of fruit ripening. Specific E2 ubiquitin-conjugating enzymes were involved in the regulation of fruit ripening by targeting transcription factor RIN (ripening inhibitor) in the tomato [54]. Liu et al. [55] displayed the interaction of banana MADS-box protein, MuMADS1, and the ubiquitin-activating enzyme, E-MuUBA, in post-harvest banana fruit. Also, the degradation of Aux/IAA, the auxin repressor, via the ubiquitin/26S proteasome pathway is possibly involved in the regulation of fruit ripening [56]. In this study, the expression of some E2 ubiquitin-conjugating enzymes and E3 ubiquitin-protein ligase enzymes genes or proteins were up-regulated while others were down-regulated during fruit ripening. In the banana, the ubiquitination-related enzymes are encoded by multigene families [9]. Considering the functional diversity of ubiquitination-related enzymes, it is not surprising to find such diversity in the expression of genes or proteins across the fruit ripening stage in this work. In addition, proteome subunit alpha type-5 has been found to be up-regulated at the protein level during fruit ripening, indicating that increased protein degradation occurred.

The disulfide bonds resulting from cysteine oxidation play important roles in the protein structure and oligomerization, as well as enzyme activity regulation. In the present study, two categories of genes/proteins involved in the regulation of disulfide bonds, protein disulfide isomerase (PDI) and thioredoxin (Trx), were differentially expressed during banana fruit ripening. PDI is an enzyme in the endoplasmic eukaryote that functions as a molecular chaperone and a folding enzyme by catalyzing the formation and breakage of the disulfide bonds of unfolded or misfolded proteins, while Trxs are mainly associated with the redox modification of protein disulfide [57]. During banana fruit ripening, the expression of several PDI and Trx genes or proteins were significantly up-regulated, indicating that the disruption of protein disulfide bonds and the oxidation of protein cysteine occurred on a large scale. The increased expression of PDI and Trx genes or proteins was beneficial to the maintenance of protein disulfide bonds and the reduction of oxidized proteins.

### 4.6. Energy Metabolism

Energy metabolism involves the release, transfer, storage, and use of energy in organisms. Glycolysis, the TCA cycle, and oxidative phosphorylation are important energy-generating pathways. In this study, most of the up-regulated and energy-related genes and proteins were associated with glycolysis. In addition, several fermentation-associated genes were also up-regulated, including alcohol dehydrogenase (ADH), pyruvate decarboxylase (PDC), and lactate dehydrogenase. During banana fruit ripening, the respiration rate showed an increasing trend (Figure 1C) while the ATP content in the peel decreased gradually (Figure 1E). Considering that glycolysis is an anaerobic process and generates only a small amount of energy stored in the form of two ATP and two NADH (Nicotinamide adenine dinucleotide) for each glucose, anaerobic respiration might predominate in the energy metabolism during banana fruit ripening. Similarly, some previous studies have also reported the increase in glycolytic and ethanol fermentation-related enzymes during fruit ripening at the transcript and protein level [48,58]. D’Ambrosio et al. [48] proposed that the enhanced glycolytic pathway provides carbon skeletons for the respiratory climax in ripe apricots. In this study, the significantly up-regulated ADH and PDC possibly contributed to the synthesis of volatile compounds [47], which is of importance to the formation of peculiar fragrance characteristics.

In aerobic metabolism, the electrons of NADH and FADH2 (reduced flavin adenine dinucleotide) produced by glucose degradation and the TCA cycle enter the electron transport chain to synthesize ATP by oxidative phosphorylation. In this study, only a few TCA-related genes or proteins (dihydrolipoyl dehydrogenase, malate dehydrogenase) were found to be up-regulated during fruit ripening. Moreover, most of the identified differentially expressed genes or proteins related to oxidative phosphorylation were down-regulated during fruit ripening, including ATP synthase F1, ATP synthase subunit alpha, ATP synthase subunit beta, NADH-ubiquinone oxidoreductase, succinate dehydrogenase subunit 3, cytochrome c oxidase subunit, and cytochrome b. Oxidative phosphorylation yields a substantially greater amount of ATP than anaerobic respiration. Our results showed reduced expression of the components of the electron transport chain and ATP synthase, which inevitably resulted in the decreased energy level. 

## 5. Conclusions

We conducted a combination of transcriptomic, proteomic, and metabolomics analyses to investigate the mechanisms underlying peel ripening in banana. Many functional genes and proteins were identified that potentially contribute to banana peel ripening, especially those associated with transcriptional regulation, hormone signaling, cell wall modification, protein modification, and energy metabolism. Specially, most of the differentially expressed transcriptional factors during ripening were ERF and bHLH families. Moreover, many auxin signaling-related genes were identified with up-regulated expression, and exogenous IAA treatment accelerated banana fruit ripening and un-regulated the expression of many auxin signaling-related and ethylene signaling-related genes, suggesting that auxin participates in the regulation of banana peel ripening. However, the interaction between ethylene and auxin signals is poorly understood and further analysis is needed to identify the interacting nodal genes. We also postulate that XTH family members play a more important role in banana peel softening. Moreover, Hsps mediated-protein modification and ubiqutin-protesome system-mediated protein degradation was involved in peel ripening. Furthermore, anaerobic respiration might predominate in energy metabolism while the efficiency of oxidative phosphorylation decreased during banana fruit ripening. Taken together, the study highlights our understanding of peel ripening in the banana, provides new clues for further dissection of specific gene functions, and contributes to the development of novel technologies to reduce postharvest losses of the fruit.

## Figures and Tables

**Figure 1 biomolecules-09-00167-f001:**
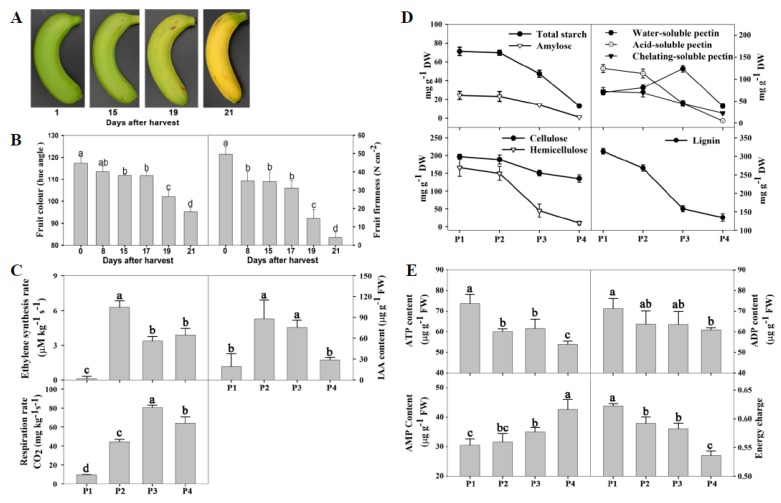
The changes in the physiological parameters of banana fruit during ripening. Banana fruit were stored at 25 °C and 85% to 90% relative humidity. (**A**) Visual appearance of banana fruit stored for 1, 15, 19, and 21 days, defined as P1, P2, P3, and P4, respectively (the same below); (**B**) fruit color, firmness, ethylene production rate, 3-indoleacetic acid (IAA) content, and respiration rate; (**C**) the contents of starch and cell wall compounds (cellulose, hemicellulose, water-soluble pectin, ionic-soluble pectin, covalent-soluble pectin, and lignin) in the peel; (**D**,**E**) the contents of adenosine triphosphate (ATP), adenosine diphosphate (ADP), and adenosine monophosphate (AMP), and energy charge in the peel. FW, fresh weight. DW, dry weight. P1, the sample at 1 day. P2, the sample at 15 days. P3, the sample at 19 days. P4, the sample at 21 days.

**Figure 2 biomolecules-09-00167-f002:**
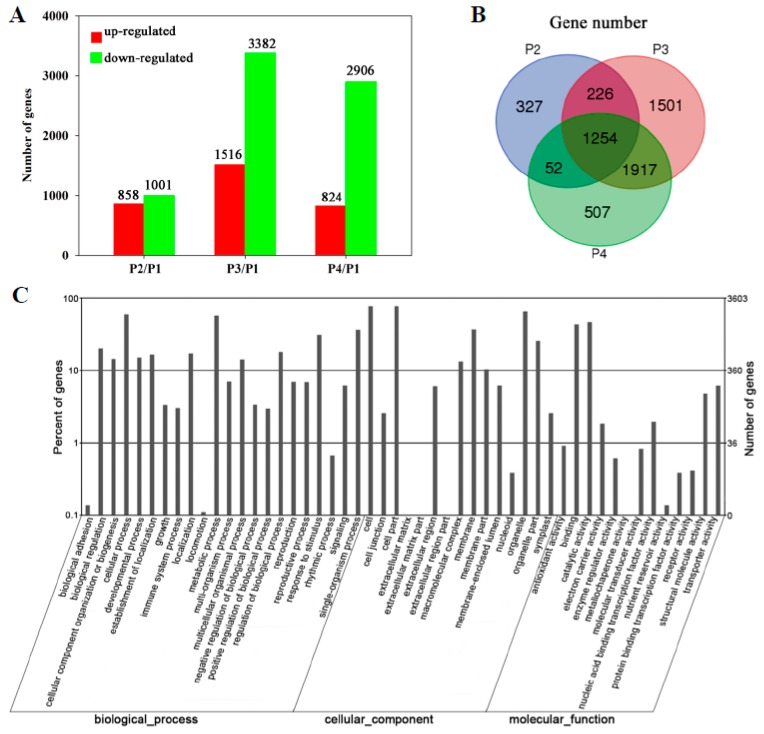
Transcriptomic profiling of banana peel during fruit ripening. (**A**) Numbers of differentially expressed genes in the peel at different stages of ripening. The numbers of up-regulated genes and down-regulated genes at different stages of ripening are indicated with red and green color, respectively; (**B**) Venn diagram illustrating the corresponding numbers of differentially expressed genes in different stages of ripening; (**C**) gene ontology term assignments based on significant plant species hits against the National Center for Biotechnology Information (NCBI) non- redundant (NR) database, according to the biological process, molecular function, and cellular component.

**Figure 3 biomolecules-09-00167-f003:**
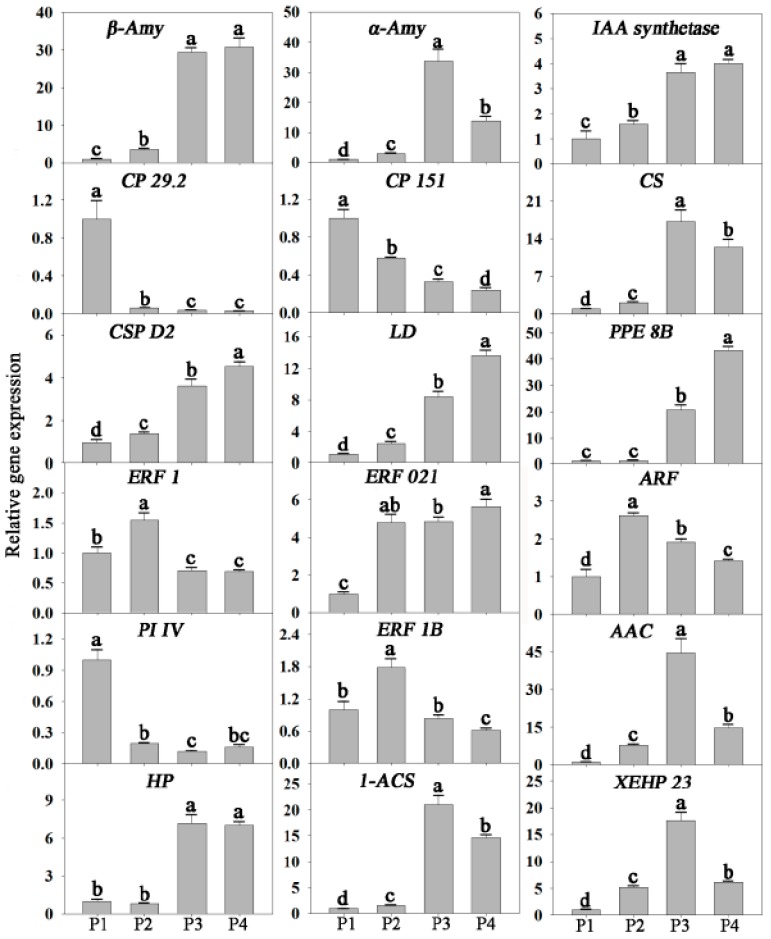
Validation of RNA-seq results by quantitative reverse transcription polymerase chain reaction (qRT-PCR). Eighteen differentially expressed genes related to signal transduction, transcription factors (TFs), photosynthesis, cell wall, starch, energy, respiration, and other metabolisms were selected for quantitative real-time PCR analysis. b-Amy, beta-amylase 3; a-Amy, alpha-amylase; GH 3.8, probable indole-3-acetic acid-amidosynthetase GH3.8; ARF, auxin-responsive family protein; ERF 021, putative ethylene-responsive transcription factor ERF021; ERF 1, putative ethylene-responsive transcription factor 1; CP 29.2, chlorophyll a-b binding protein CP29.2, chloroplastic; CP 151, chlorophyll a-b binding protein 151, chloroplastic; CS, citrate synthase; AAC, ADP, ATP carrier protein, mitochondrial; ERF 1B, ethylene-responsive transcription factor 1B; PI IV, Photosystem I reaction center subunit IV, chloroplastic; CSP D2, cellulose synthase-like protein D2; LD, Leucoanthocyanidin dioxygenase; PPE 8B, pectinesterase/pectinesterase inhibitor PPE8B; XEHP 23, xyloglucan endotransglucosylase/hydrolase protein 23; 1-ACS, 1-aminocyclopropane-1-carboxylate synthase; HP, Hypothetical protein. Values with different letters are significantly different (*p* < 0.05).

**Figure 4 biomolecules-09-00167-f004:**
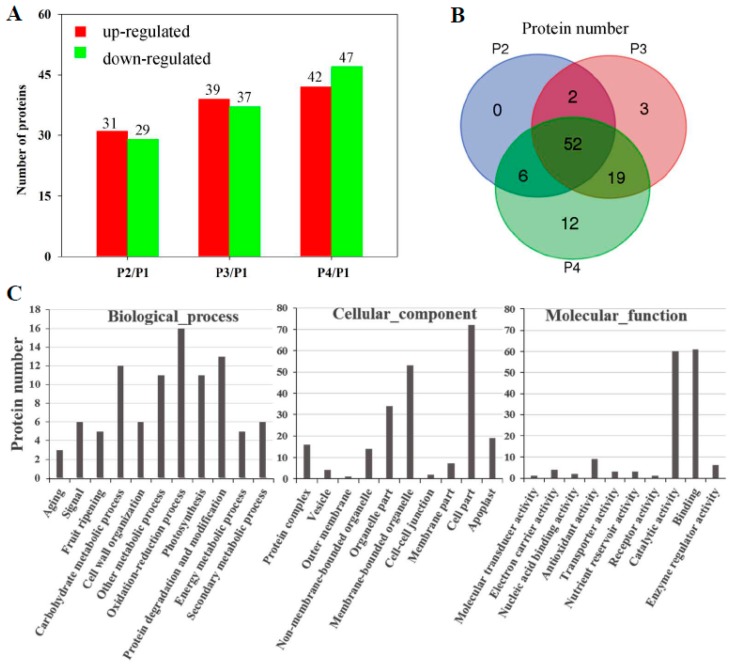
Differentially expressed proteins in the peel of banana fruit at different stages of ripening. Comparative proteomics of banana peel were performed using 2-DE coupled matrix-assisted laser desorption/ionization time-of-flight tandem mass spectrometry (MALDI-TOF MS/MS). (**A**) Numbers of differentially expressed proteins. The numbers of up-regulated proteins and down-regulated proteins are indicated with red and green color, respectively; (**B**) Venn diagram illustrating the corresponding numbers of differentially expressed proteins at different stages of ripening; (**C**) gene ontology term assignments based on significant plant species hits against the NR database, according to the biological process, molecular function, and cellular component.

**Figure 5 biomolecules-09-00167-f005:**
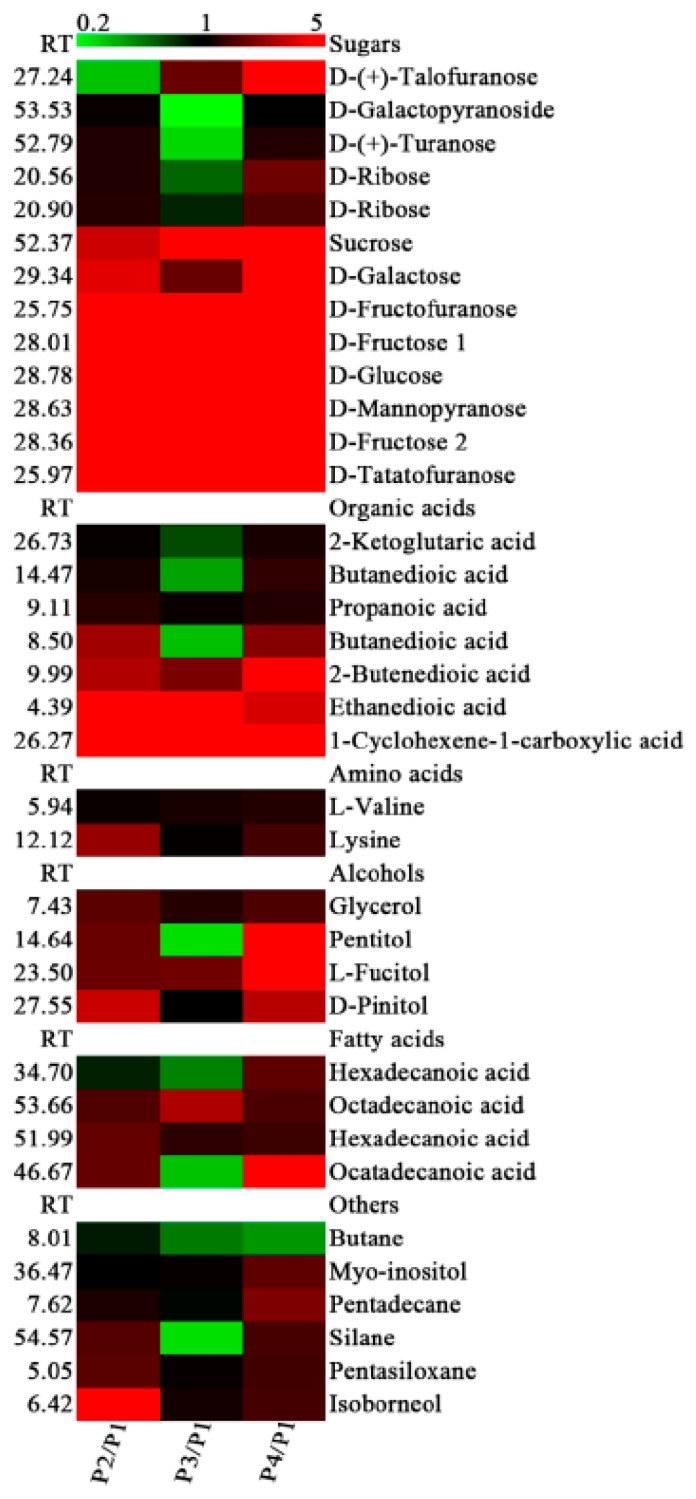
Differentially accumulated primary metabolites in the peel of banana fruit at different stages of ripening. The ratios of P2/P1, P3/P1, and P4/P1 are shown. The red squares and the green squares indicate up-regulation and down-regulation of metabolites, respectively.

**Figure 6 biomolecules-09-00167-f006:**
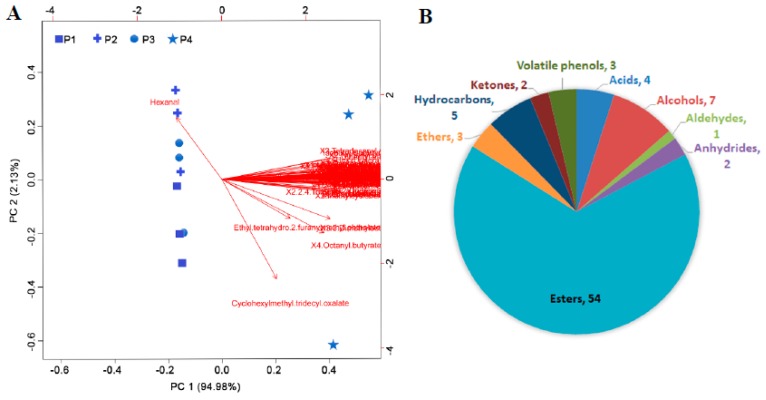
(**A**) Principal component analysis of differentially accumulated volatile compounds in the peel of banana fruit at different stages of ripening. All those compounds were specifically accumulated at P4 except one aldehyde. Detailed information is shown in Appendix A. (**B**) Classification of differentially accumulated volatile compounds in the peel of banana fruit at different stages of ripening.

**Figure 7 biomolecules-09-00167-f007:**
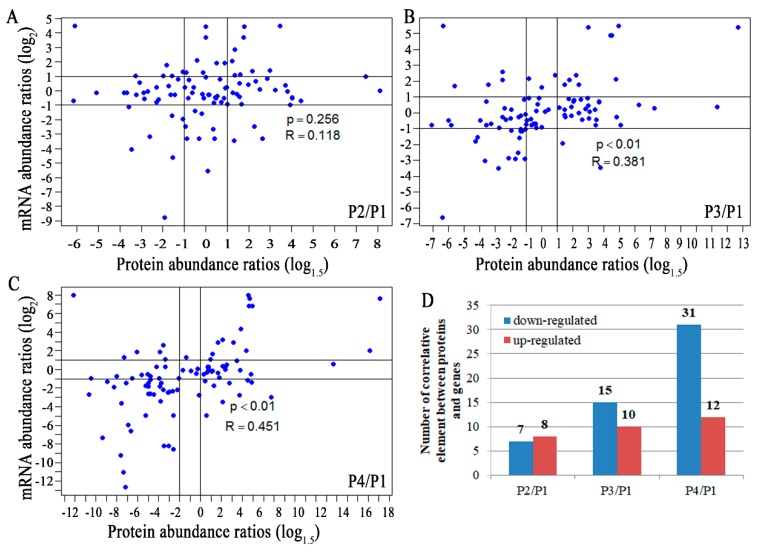
mRNA abundance ratios versus protein-abundance ratios. Ninety-four differentially expressed proteins and their corresponding genes were analyzed. The log 2 ratio of P2, P3, P4 to P1 at the mRNA level and the log1.5 ratio at the protein level were plotted against each other for each unique gene. The number of the correlative element between proteins and genes was counted and shown in the **A**, **B**, and **C** histogram. **D**, statistical count of up- or down-regulated elements between proteins and genes.

**Figure 8 biomolecules-09-00167-f008:**
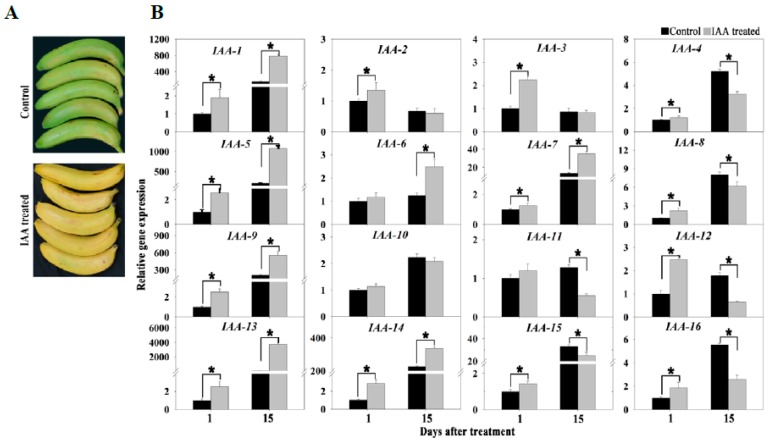
Effect of exogenous IAA treatment on fruit ripening and gene expression. (**A**) Visual appearance of banana fruit treated with 0.1 mM IAA after 15 d of storage. (**B**) Expression of selected ripening-related genes in the peel of banana fruit with 0.1 mM IAA after 1 and 15 days of storage. IAA-1, auxin-responsive family protein; IAA-2, OsSAUR57-auxin-responsive SAUR gene family member; IAA-3, auxin-induced protein 22D; IAA-4, OsSAUR37 - auxin-responsive SAUR gene family member; IAA-5, auxin-responsive family protein; IAA-6, SAUR family protein; IAA-7, 1-aminocyclopropane-1-carboxylate oxidase; IAA-8, ethylene-responsive transcription factor ERF024; IAA-9, 1-aminocyclopropane-1-carboxylate synthase; IAA-10, EIN3-binding F-box protein 1; IAA-11, ethylene receptor; IAA-12, ethylene-responsive transcription factor 1B; IAA-13, xyloglucan endotransglucosylase/hydrolase protein 32; IAA-14, polygalacturonase; IAA-15, pectinesterase/pectinesterase inhibitor 12; IAA-16, probable pectinesterase 53. Asterisk represents the statistical differences (*p* < 0.05) between control fruit and IAA-treated fruit.

**Figure 9 biomolecules-09-00167-f009:**
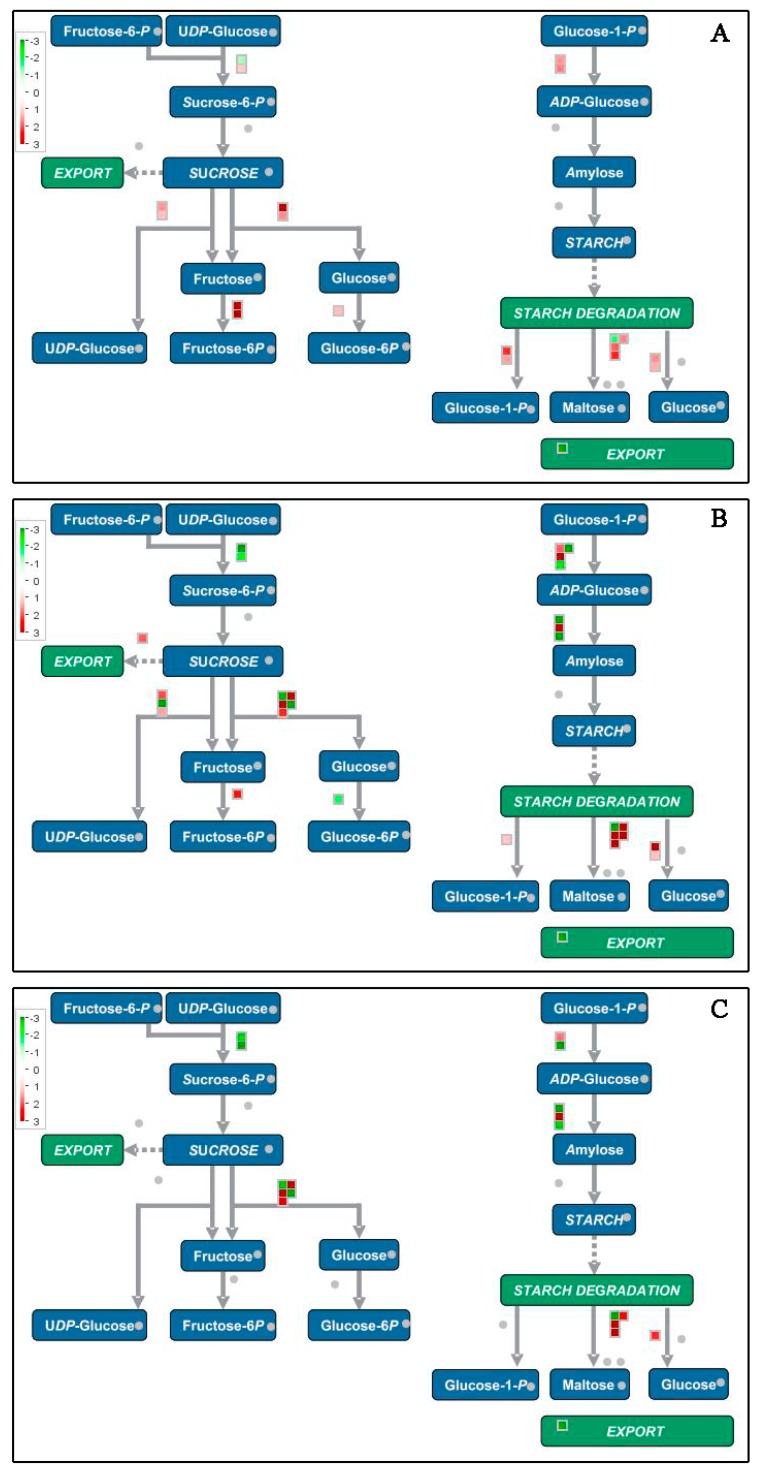
Schematic diagram of starch and sucrose metabolism using the MapMan visualization platform. The gene expression ratios of P2 (**A**), P3 (**B**), and P4 (**C**) compared with P1 were used in MapMan analysis. The red squares and green squares indicate up- or down-regulated genes involved in starch and sucrose metabolism, respectively.

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
