# Peer review of "Integrated Transcriptomic, Proteomic, and Metabolomics Analysis Reveals Peel Ripening of Harvested Banana under Natural Condition"

_biomolecules, 2019, doi:10.3390/biom9050167_

Round 1
Reviewer 1 Report
The paper presented by Yun et al. is interesting, reports new data on peel ripening of harvested banana under natural condition, and should be published in Biomolecules after minor revision.
Comments for the author:
Regarding to this work I have some comments:
- In Materials and Methods section, Page 4, Line 45: Please describe briefly the procedure to collect and detect volatile compounds. The method described in Jing et al. (2015) is for papaya fruits.
- In Figure 5: D-Fructose is reported as D-Fructose 1 and D-Fructose 2 in metabolomics profiling; please indicate the meaning of this designation.
- In Results section: For the PCA analysis (Figure 6), it would be interesting to include a Supplementary Table with metabolite contributions to the principal components (especially for PC1), or mention in the text which were the volatile compounds that most contribute to PC1 separation (Figure 6A).
Minor remarks:
- Page 4, Line 18: change “sports” by “spots”
- Page 6, Lines 9 and 10: firmness must be expressed in N cm-2 instead of N cm-1
- Page 10, Line 4: “Metabolomic profiling” should be “Metabolomics profiling”
- Page 18, Line 28: “…in involved degradation…” should be “…in involved in the degradation…”
-Page 20, Line19: the term “respiration rate” is twice, please correct this
Author Response
Reviewer’s comment: The paper presented by Yun et al. is interesting, reports new data on peel ripening of harvested banana under natural condition, and should be published in Biomolecules after minor revision.
Authors’ response: We thank the reviewer very much for the positive comment.
Reviewer’s comment: Comments for the author:
Regarding to this work I have some comments:
- In Materials and Methods section, Page 4, Line 45: Please describe briefly the procedure to collect and detect volatile compounds. The method described in Jing et al. (2015) is for papaya fruits.
Authors’ response: We have provided detailed information on the procedure to collect and detect volatile compounds.
Reviewer’s comment: - In Figure 5: D-Fructose is reported as D-Fructose 1 and D-Fructose 2 in metabolomics profiling; please indicate the meaning of this designation.
Authors’ response: Yes, D-fructose had two peaks in the spectrum when we detect it by GC-MS, so we named them D-Fructose 1 and D-Fructose 2.
Reviewer’s comment: - In Results section: For the PCA analysis (Figure 6), it would be interesting to include a Supplementary Table with metabolite contributions to the principal components (especially for PC1), or mention in the text which were the volatile compounds that most contribute to PC1 separation (Figure 6A).
Authors’ response: We have added PCI and PC2 values for each compound in Supplementary Table S1. Please find the Supplementary Table S1 for details.
Reviewer’s comment: Minor remarks:
- Page 4, Line 18: change “sports” by “spots”
- Page 6, Lines 9 and 10: firmness must be expressed in N cm-2 instead of N cm-1
- Page 10, Line 4: “Metabolomic profiling” should be “Metabolomics profiling”
- Page 18, Line 28: “…in involved degradation…” should be “…in involved in the degradation…”
-Page 20, Line19: the term “respiration rate” is twice, please correct this
Authors’ response: According to these suggestions, we have made corresponding revision.
Reviewer 2 Report
The manuscript by Yun and co-workers reports about a comprehensive transcriptomic, proteomic and metabolomics analysis of peel ripening mechanism in banana. To study the time-dependent ripening process, banana peel samples were analysed from initial mature green stage (110 days after anthesis) and 1, 8, 15, 17, 19 and 21 days after harvest. The author’s systematic approach included optical and physical characterisation of banana peel firmness, chemical analysis of ethylene production rate, IAA content, ATP, ADP, AMP, and cell wall composition. Further peel analysis included RNA sequencing (and qPCR for RNA data validation), protein analysis by gel electrophoresis followed MADLI mass spectrometry, and metabolic profiling by GC-MS. The results are discussed as separate sections on physiological characteristics and transcriptomic, proteomic, and metabolomic analysis and represented in nine figures and supporting information.
The manuscript is clearly and systematically structured to represent and discuss the comprehensive characterisation data. The manuscript provides an in-depth analysis shining light into the complex mechanism and its many involved process of banana peel ripening. This reviewer thinks that the manuscript is an interesting and valuable contribution for the studies of fruit quality and ripening physiology. As such, this reviewer recommends the manuscript for publication in Molecules. Additional minor comments to the manuscript are listed below.
General comment:
1) Discussion: Please add a statement about the limitation of current work and what studies should be performed in the future to further deepen the understanding of the banana ripening process. This could also be discussed briefly in the concluding section.
Specific comments:
P. 3, line 2: “IAA” abbreviation is not defined. Please define abbreviation at first occurrence.
Figure 7: To improve the clarity of Figure 7, this reviewer recommends to add labels (A), (B), (C), (D) to the four graphs.
Author Response
Reviewer’s comment: Comments and Suggestions for Authors
The manuscript by Yun and co-workers reports about a comprehensive transcriptomic, proteomic and metabolomics analysis of peel ripening mechanism in banana. To study the time-dependent ripening process, banana peel samples were analysed from initial mature green stage (110 days after anthesis) and 1, 8, 15, 17, 19 and 21 days after harvest. The author’s systematic approach included optical and physical characterisation of banana peel firmness, chemical analysis of ethylene production rate, IAA content, ATP, ADP, AMP, and cell wall composition. Further peel analysis included RNA sequencing (and qPCR for RNA data validation), protein analysis by gel electrophoresis followed MADLI mass spectrometry, and metabolic profiling by GC-MS. The results are discussed as separate sections on physiological characteristics and transcriptomic, proteomic, and metabolomic analysis and represented in nine figures and supporting information.
The manuscript is clearly and systematically structured to represent and discuss the comprehensive characterisation data. The manuscript provides an in-depth analysis shining light into the complex mechanism and its many involved process of banana peel ripening. This reviewer thinks that the manuscript is an interesting and valuable contribution for the studies of fruit quality and ripening physiology. As such, this reviewer recommends the manuscript for publication in Molecules. Additional minor comments to the manuscript are listed below.
Authors’ response: We thank the reviewer very much for the positive comment.
Reviewer’s comment: General comment:
1) Discussion: Please add a statement about the limitation of current work and what studies should be performed in the future to further deepen the understanding of the banana ripening process. This could also be discussed briefly in the concluding section.
Authors’ response: According to the suggestion, we have made corresponding revision. Both ethylene and auxin signaling played an important role in peel ripening of harvested banana fruit. There might be cross-talk between ethylene signaling and auxin signaling. However, the interaction between ethylene and auxin signals is poorly understood and further analysis is needed to identify the interacting nodal genes. We speculated that bZIPs and NACs might play an important role in the synthesis of volatile compounds in banana peel. However, the transcription factors that regulate the synthesis and accumulation of aromatic esters have not been confirmed, and further analysis is needed to verify their gene functions. Detail information was shown in Page 20 Lines 12-14, Page 22 Lines 19-21, and Page 24 Lines 13-15.
Reviewer’s comment: Specific comments:
P. 3, line 2: “IAA” abbreviation is not defined. Please define abbreviation at first occurrence.
Figure 7: To improve the clarity of Figure 7, this reviewer recommends to add labels (A), (B), (C), (D) to the four graphs.
Authors’ response: According to these suggestions, we have made corresponding revision.